# DREAMER XL: TOWARDS HIGH-RESOLUTION TEXT-TO-3D GENERATION VIA TRAJECTORY SCORE MATCHING

## ABSTRACT

In this work, we propose a novel Trajectory Score Matching (TSM) method that aims to solve the pseudo ground truth inconsistency problem caused by the accumulated error in Interval Score Matching (ISM) when using the Denoising Diffusion Implicit Models (DDIM) inversion process. Unlike ISM which adopts the inversion process of DDIM to calculate on a single path, our TSM method leverages the inversion process of DDIM to generate two paths from the same starting point for calculation. Since both paths start from the same starting point, TSM can reduce the accumulated error compared to ISM, thus alleviating the problem of pseudo ground truth inconsistency. TSM enhances the stability and consistency of the model's generated paths during the distillation process. We demonstrate this experimentally and further show that ISM is a special case of TSM. Furthermore, to optimize the current multi-stage optimization process from high-resolution text to 3D generation, we adopt Stable Diffusion XL for guidance. In response to the issues of abnormal replication and splitting caused by unstable gradients during the 3D Gaussian splatting process when using Stable Diffusion XL, we propose a pixel-by-pixel gradient clipping method. Extensive experiments show that our model significantly surpasses the state-of-the-art models in terms of visual quality and performance.

## 1 INTRODUCTION

In recent years, Virtual Reality (VR) and Augmented Reality (AR) have increasingly become a part of our daily lives, and the demand for high-quality 3D content has increased significantly. 3D technology has become extremely important, allowing us to visualize, understand and interact with complex objects and environments. It also plays a key role in various fields such as architecture, animation, gaming and virtual reality. In addition, 3D technology shows broad application prospects in retail (Wodehouse & Abba, 2016), online meetings (Nakanishi et al., 1999), education (Reisoğlu et al., 2017) and other fields (Miao et al., 2024). Despite its wide application, the complexity of creating 3D content poses considerable challenges: generating high-quality 3D models requires computional time, effort, and expertise. Given these challenges, methods for generating 3D from text have become particularly important in recent years (Lin et al., 2023; Zhu & Zhuang, 2023; Ma et al., 2023; Shi et al., 2023b). These methods create accurate 3D models directly from natural language descriptions, thereby reducing manual input in traditional 3D modeling processes. Once the text-to-3D method can efficiently generate large amounts of data, it will not only shorten the production time of 3D content, but also reduce costs and improve production efficiency.

Typically, text-to-3D generation methods utilize pre-trained text-to-image diffusion models (Saharia et al., 2022) as an image prior to training neural parametric 3D models such as Neural Radiance Fields (NeRF) (Mildenhall et al., 2021) and 3D Gaussian splitting (Kerbl et al., 2023). These approaches enable the rendering of consistent images that are aligned with the text. This process essentially relies on Score Distillation Sampling (SDS) (Poole et al., 2022). Through SDS, the model can distill the capabilities of

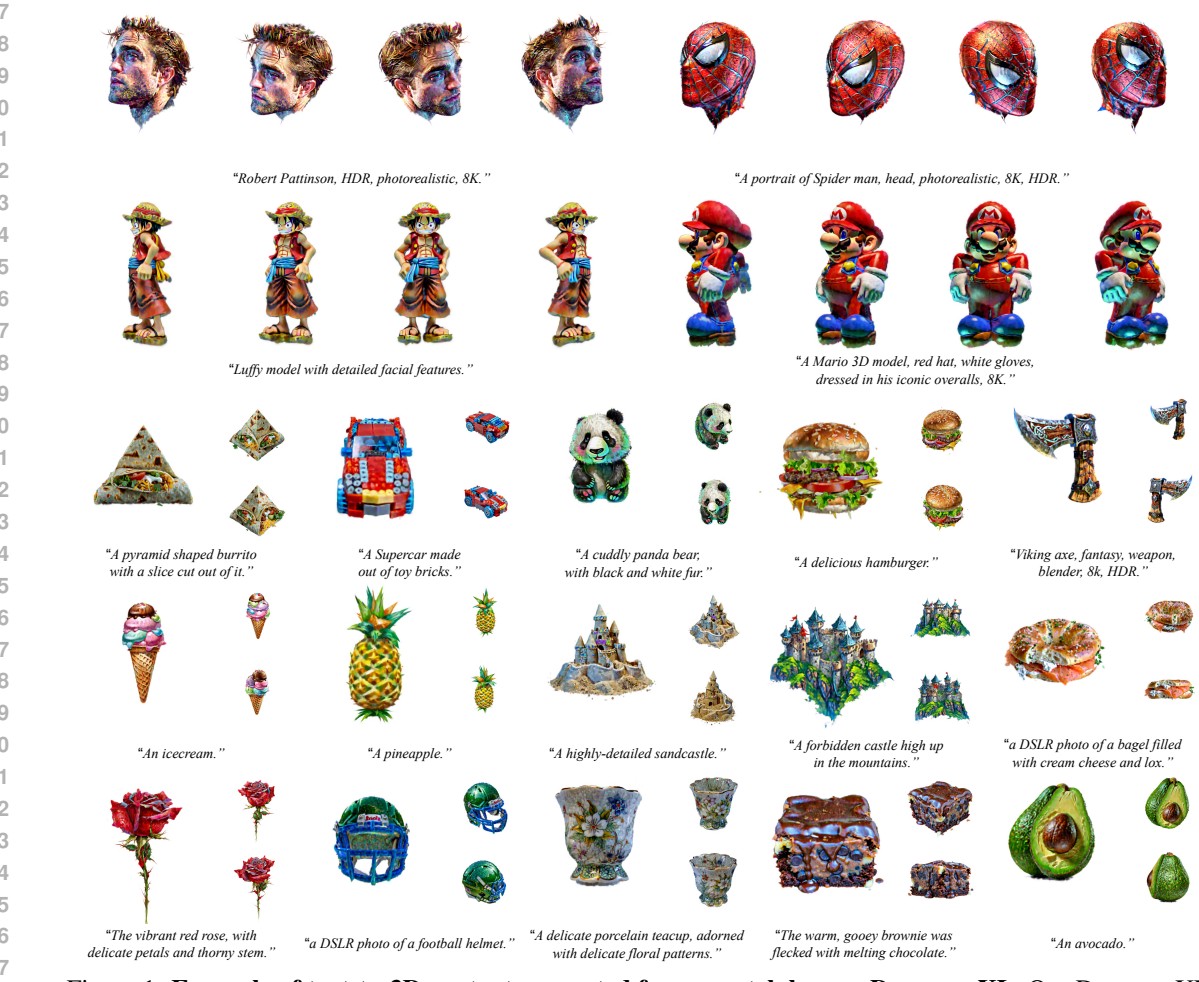

"Robert Pattinson, HDR, photorealistic, 8K."

"A portrait of Spider man, head, photorealistic, 8K, HDR."

"Luffy model with detailed facial features."

"A Mario 3D model, red hat, white gloves, dressed in his iconic overalls, 8K."

"A pyramid shaped burrito with a slice cut out of it."

"A Supercar made out of toy bricks."

"A cuddly panda bear, with black and white fur."

"A delicious hamburger."

"Viking axe, fantasy, weapon, blender, 8k, HDR."

"An icecream."

"A pineapple."

"A highly-detailed sandcastle."

"A forbidden castle high up in the mountains."

"a DSLR photo of a bagel filled with cream cheese and lox."

"The vibrant red rose, with delicate petals and thorny stem."

"a DSLR photo of a football helmet."

"A delicate porcelain teacup, adorned with delicate floral patterns."

"The warm, gooey brownie was flecked with melting chocolate."

"An avocado."

Figure 1: **Example of text-to-3D content generated from scratch by our Dreamer XL.** Our Dreamer XL is based on 3D Gaussian splatting using stable diffusion XL. Please zoom in for details.

the pre-trained 2D diffusion model to obtain rendered images, and optimize the parameters of the 3D model through backpropagation, so that the 3D model can be effectively trained even without actual image data. However, since random noise generates inconsistent pseudo-baselines, the results obtained by SDS optimization of 3D models tend to be averaged, leading to problems such as over-smoothing.

Although some recent work (Wang et al., 2024; Liang et al., 2023) devote themselves to solving the over-smoothing problems, they inevitably lead to the generation of low-resolution and average results due to the inherent limitations of stable diffusion models and their sampling methods. For example, (Liang et al., 2023), inspired by the DDIM inversion process, proposed interval score matching, which can generate relatively consistent results. However, due to the inherent cumulative errors in the DDIM inversion process, it may lead to the averaging of results in certain regions. Furthermore, most existing methods do not yet support the new high-resolution Stable Diffusion XL (SDXL) (Podell et al., 2023; Luo et al., 2023). To achieve an output of 1024x1024 high resolution, multi-stage optimization is necessary. The main reason is the inherent instability of the Variational Autoencoder (VAE) in the SDXL (Podell et al., 2023) architecture, which is particularly evident during the optimization process of the 3D Gaussian Splatting. In this process, the gradients directly

affect the duplication and deletion of point clouds in 3D space. The anomalous gradients introduced by SDXL severely hinder the optimization process of 3D Gaussian Splatting, leading to generated 3D models that lose complex texture details, have blurred appearances, and exhibit abnormal colors. In severe cases, this can cause the 3D models to fail to converge.

In this work, we aim to overcome the above limitations. The reverse process of DDIM was adopted by (Liang et al., 2023), effectively reducing the higher reconstruction errors generated by the diffusion model's one-step reconstruction and generating relatively consistent pseudo-ground truth. However, the inherent accumulation errors in the reverse process of DDIM still result in semantic changes in these pseudo-ground truths, leading to partially averaged reconstruction results in certain regions, resulting in erroneous and unrealistic outcomes in these regions. To address this issue, we propose a novel method called Trajectory Score Matching (TSM). We through simple improvements to Interval Score Matching (ISM) (Liang et al., 2023), effectively alleviate the average effect of inconsistent pseudo-ground truths caused by inherent accumulation errors. We demonstrate that our TSM has smaller accumulation errors compared to ISM, and ISM can be considered as a special case of TSM. Through experiments, we prove that the effects produced by our TSM are superior to ISM, yielding highly realistic and detailed results. In order to achieve high-resolution output, previous methods need to undergo multi-stage training. Our model directly uses SDXL that supports high resolution as guidance without going through multi-stage training. This not only reduces training costs, but also simplifies the training process. However, we still need to solve the inherent gradient instability problem of SDXL. Therefore, we propose a pixel-by-pixel gradient clipping method, which effectively alleviates the inherent gradient instability of SDXL. In summary, the contributions of our work are as follows:

- We investigate and analyze the inherent accumulated errors produced by DDIM inversion process in interval score matching (ISM), resulting in the presence of inconsistent pseudo-ground truth.

- To address the aforementioned limitation, we introduce a novel Trajectory Score Matching (TSM) method. Unlike the single path of ISM, TSM improves ISM to a dual path, effectively alleviating the inconsistent pseudo-ground truth issue generated from the inherent accumulated error of DDIM.

- To simplify the training process and generate high-resolution, high-quality text-to-3D results. We are the first to leverage SDXL for guidance based on 3D Gaussian splatting. In addition, we also introduce a novel gradient clipping method, which effectively solves the problem of SDXL in gradient stability. Extensive experiments demonstrate that our method significantly outperforms the current state-of-the-art methods.

## 2 RELATED WORK

**2D diffusion.** Score-based generative models and diffusion models (Song & Ermon, 2019; Song et al., 2020b; Ho et al., 2020; Balaji et al., 2022; Saharia et al., 2022; Podell et al., 2023) have shown excellent performance in image synthesis (Dhariwal & Nichol, 2021), especially by introducing latent diffusion models (LDM) (Rombach et al., 2022) into stable diffusion to generate high-resolution images in latent space. Podel et al. (Podell et al., 2023) further extended this model to a larger latent space in SDXL, VAE and U-net, achieving higher resolution (1024×1024). Zhang et al. (Zhang et al., 2023) enhanced the functionality of these models by generating controllable images for different input types. At the same time, the diffusion model also showed impressive performance in converting text into image synthesis, which opens up the possibility of using this technology to directly generate 3D images from text (Chen et al., 2023; Hong et al., 2022; Lin et al., 2023; Michel et al., 2022; Poole et al., 2022; Wang et al., 2024).

**Text-to-3D Generation.** Early attempts from text-to-3D are mainly guided by the use of multi-modal information from CLIP (Radford et al., 2021) to achieve information conversion from text-to-3D, with DreamField (Jain et al., 2021) being a pioneer in this direction. However, the multi-modal information of CLIP can only provide rough alignment, and the results of using it for 3D distillation are often unsatisfactory.

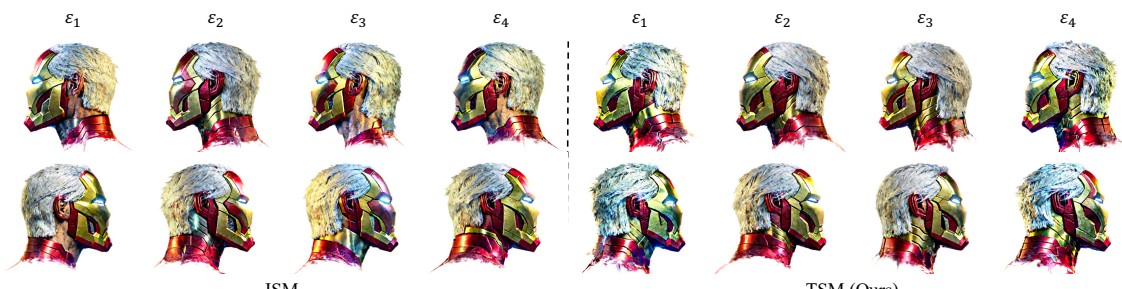

Figure 2: **ISM and our TSM example (Liang et al., 2023).** We observed that when using the same initial $x_0$, timestep $t$, and fixed noise $\epsilon$, the results generated by ISM exhibit significant inconsistencies (demonstrated by four different noise generation outcomes). These inconsistencies arise from the inherent accumulation of errors during the DDIM inversion process, leading to deviations in the approximation $\hat{x}_0$. In contrast, TSM mitigates this issue, resulting in more consistent features and styles across different viewpoints.

Zero-1-to-3 (Liu et al., 2023b) first introduces camera parameters to fine-tune the 2D pre-trained stable diffusion model, enabling it to generate multi-view images, and then use the multi-view images for 3D reconstruction. This improvement not only facilitates more accurate 3D reconstructions, but also inspires a wealth of derivative research (Metzer et al., 2023; Liu et al., 2024; Qian et al., 2023; Liu et al., 2023a; Shi et al., 2023a). In addition, another research direction explores the possibility of using pre-trained 2D diffusion models to directly optimize 3D representations. These methods often combine differentiable 3D representation techniques, such as NeRF (Mildenhall et al., 2021), NeuS (Wang et al., 2021), and 3D Gaussian Splatting (Kerbl et al., 2023), and optimize model parameters through backpropagation techniques. Dreamfusion (Poole et al., 2022) first introduces SDS to optimize 3D representations directly from pre-trained 2D text-to-image diffusion models. Similarly, Score Jacobian Chaining (Wang et al., 2023) proposes an alternative method that achieves parameterization effects similar to SDS. ProlificDreamer (Wang et al., 2024) conducted an in-depth analysis of the objective function of SDS and proposed a particle-based variational framework called Variational Score Distillation (VSD), which significantly improves the quality of generated content. The latest research combines SDS with Gaussian Splatting to accelerate the optimization process. Consistent3D (Wu et al., 2024) analyzes SDS from the latest perspective of ordinary differential equations (ODE) and proposes a method called Consistency Distillation Sampling (CSD) to solve the challenges of SDS in over-smoothing and inconsistency issues. Similarly, LucidDreamer (Liang et al., 2023) analyzed the loss function of SDS and proposed interval score matching (ISM), which is very similar to the idea of CSD. However, ISM utilizes the reversible diffusion trajectory of DDIM (Song et al., 2020a) when calculating the two interval steps. DDIM will inevitably produce inherent accumulated errors in this process, resulting in inconsistent reconstruction results. In this work, we empirically follow the mature mainstream architecture method of 3D Gaussian Splatting (Liang et al., 2023; Tang et al., 2023; Yi et al., 2024) as the baseline of our approach. On this basis, inspired by the recent Consistency trajectory model (Kim et al., 2023), we propose to optimize the 3D model from two trajectories, and use the less noisy trajectory to guide another noisier trajectory to alleviate the inconsistency problem.

## 3 METHODS

This section presents the preliminaries on the inverse DDIM process and ISM (see Section 3.1). We then propose the Trajectory Score Matching (TSM) method (see Section 3.2), which generates dual paths from the same starting point using the reverse process of DDIM. This enhances the stability and consistency of the model along the entire generative path during the distillation process. We further indicate that ISM is a

special case of TSM (see Section 3.3). Additionally, we investigate the challenges of optimizing 3D models using the SDXL architecture (see Section 3.4).

## 3.1 PRELIMINARIES

**Review of DDIM inversion** We first consider the most common sampling scheme is that of DDIM (Song et al., 2020a) where intermediate steps are calculated as:

$$x_{t-1} = \sqrt{\alpha_{t-1}} \left( \frac{x_t - \sqrt{1 - \alpha_t} \epsilon_\phi(x_t, t, \emptyset)}{\sqrt{\alpha_t}} \right) + \sqrt{1 - \alpha_{t-1}} \epsilon_\phi(x_t, t, \emptyset), \tag{1}$$

where $x_t$ and $x_{t-1}$ represent the noisy latent, $\{\alpha_t\}_{t=0}^T$ (where $\alpha_0 = 1, \alpha_T = 0$) indicates a set of time steps indexing a strictly monotonically reducing noise schedule. The $\epsilon_\phi(x_t, t, \emptyset)$ denotes the predicted denoising direction (by stable diffusion model) with the given condition $y$ (The condition is null, *i.e.*, unconditioned $\emptyset$).

The above denoising process can approximate the inverse transition from $x_t$ to $x_{t-1}$, which is:

$$\begin{aligned} x_t &= \sqrt{\alpha_t} \left( \frac{x_{t-1} - \sqrt{1 - \alpha_t} \epsilon_\phi(x_t, t, \emptyset)}{\sqrt{\alpha_{t-1}}} \right) + \sqrt{1 - \alpha_t} \epsilon_\phi(x_t, t, \emptyset) \\ &\approx \sqrt{\alpha_t} \left( \frac{x_{t-1} - \sqrt{1 - \alpha_t} \epsilon_\phi(x_{t-1}, t - 1, \emptyset)}{\sqrt{\alpha_{t-1}}} \right) + \sqrt{1 - \alpha_t} \epsilon_\phi(x_{t-1}, t - 1, \emptyset), \end{aligned} \tag{2}$$

where the approximation is a linearization assumption that $\epsilon_\phi(x_t, t, \emptyset) \approx \epsilon_\phi(x_{t-1}, t - 1, \emptyset)$. This approximation inevitably introduces errors, resulting in inconsistencies between the diffusion states in the forward and backward processes.

**Text-to-3D generation by interval score matching (ISM)** The concepts of the ISM is first introduced by LucidDreamer (Liang et al., 2023) to address the issues of over-smoothness and inconsistency inherent in the original SDS method. The 3D model leverages a differentiable function $x = g(\theta, c)$ to render images, where $\theta$ represents the trainable 3D parameters and $c$ is camera parameter. The gradient of the ISM loss for $\theta$ is expressed as follows:

$$\nabla_\theta \mathcal{L}_{\text{ISM}}(\theta) := \mathbb{E}_{t,c} \left[ \omega(t)(\epsilon_\phi(x_t, t, y) - \epsilon_\phi(x_s, s, \emptyset)) \frac{\partial x}{\partial \theta} \right], \tag{3}$$

where $0 < s < t$, the noisy latent $x_t$ and $x_s$ are calculated by DDIM inversion process, the $\epsilon_\phi(x_t, t, y)$ is the predicted denoising direction given the conditioned $y$, and $\omega(t)$ is a time-dependent weighting function. ISM employs DDIM inversion during the noise addition process in optimization to mitigate the inconsistencies between pseudo-ground truths caused by random noise in SDS. Specifically, ISM adds noise through reverse iterations, progressively adding noise to the unknown $x_0$ up to a certain timestep $t_i$, resulting in $x_{t_i}$. It then computes an approximation $\hat{x}_0$ of $x_{t_i}$ and adds noise stepwise until it reaches $x_t$. However, during the DDIM inversion process, the accumulated error is inevitable, leading to discrepancies between each $\hat{x}_0$ approximated from different timesteps $t$ and the original $x_0$. These errors are further amplified when conditional denoising is introduced during the iterative process. According to recent work (Yu et al., 2023), the gradients of SDS-like methods can be decomposed into a reconstruction term and classifier-free guidance terms. The primary improvement of ISM focuses on the reconstruction term, which can be expressed as:

$$\delta_{\text{recon}} := \epsilon_\phi(x_s, s, \emptyset) - \epsilon_\phi(x_t, t, \emptyset), \tag{4}$$

However, due to the nature of DDIM inversion, $x_s$ and $x_t$ originate from different approximations of $\hat{x}_0$, meaning $x_s$ is derived from $\hat{x}_0^{s-1}$ and $x_t$ from $\hat{x}_0^s$. Additionally, because the DDIM sampling process is deterministic, it generates two distinct trajectories. Therefore, inconsistent pseudo-ground truths will be generated when optimizing the 3D model, thus affecting the optimization quality of the final result, as shown in Figure 2.

**Algorithm 1** Trajectory Score Matching

1: Initialization: DDIM inversion step size $\delta_T$ and $\delta_S$, the target prompt $y$, the offset rate $\gamma \in [0, 1]$
2: **while** $\theta$ is not converged **do**
3:     Sample: $x_0 = g(\theta, c), t \sim \mathcal{U}(1, 1000)$
4:     let $s = t - \delta_T$, $n = s/\delta_S$, and $\mu = s + \gamma\delta_T$
5:     **for** $i = [0, ..., n-1]$ **do**
6:         $x_{(i+1)\delta_S} = \sqrt{\alpha_{(i+1)\delta_S}}\left(\frac{x_{i\delta_S} - \sqrt{1-\alpha_{i\delta_S}}\epsilon_\phi(x_{i\delta_S}, i\delta_S, \emptyset)}{\sqrt{\alpha_{i\delta_S}}}\right) + \sqrt{1-\alpha_{(i+1)\delta_S}}\epsilon_\phi(x_{i\delta_S}, i\delta_S, \emptyset)$
7:     **end for**
8:     predict $\epsilon_\phi(x_s, s, \emptyset)$, then step $x_s \to x_t$ and $x_s \to x_\mu$ via
        $x_t = \sqrt{\alpha_t}\left(\frac{x_s - \sqrt{1-\alpha_s}\epsilon_\phi(x_s, s, \emptyset)}{\sqrt{\alpha_s}}\right) + \sqrt{1-\alpha_t}\epsilon_\phi(x_s, s, \emptyset)$
        $x_\mu = \sqrt{\alpha_\mu}\left(\frac{x_s - \sqrt{1-\alpha_s}\epsilon_\phi(x_s, s, \emptyset)}{\sqrt{\alpha_s}}\right) + \sqrt{1-\alpha_\mu}\epsilon_\phi(x_s, s, \emptyset)$
9:     predict $\epsilon_\phi(x_t, t, y)$, $\epsilon_\phi(x_\mu, \mu, \emptyset)$ and compute TSM gradient
        $\nabla_\theta L_{\text{TSM}} = \omega(t)(\epsilon_\phi(x_t, t, y) - \epsilon_\phi(x_\mu, \mu, \emptyset))$
10:   update $x_0$ with $\nabla_\theta L_{\text{TSM}}$
11: **end while**

## 3.2 TRAJECTORY SCORE MATCHING

To alleviate the inherent accumulated error in the DDIM inversion process, which leads to the production of inconsistent pseudo-ground truths and consequently suboptimal 3D models. Inspired by recent work (Kim et al., 2023), we propose a new approach, Trajectory Score Matching (TSM), which utilizes dual paths originating from the same starting point to minimize error accumulation during iterations. Specifically, similar to ISM (Liang et al., 2023), our TSM also utilizes the DDIM inversion to predict an invertible noisy latent trajectory. For a given timestep $s$ (where $0 < s < t$), the corresponding noise latent $x_s$ can be obtained using Equation (2). Considering $x_s$ as the starting latent, it is possible to approximate two noise latents, $x_\mu$ and $x_t$, on the latent trajectory, where $0 < s < \mu < t \le T$. This can be expressed as follows:

$$x_\mu = \sqrt{\alpha_\mu}\left(\frac{x_s - \sqrt{1-\alpha_s}\epsilon_\phi(x_s, s, \emptyset)}{\sqrt{\alpha_s}}\right) + \sqrt{1-\alpha_\mu}\epsilon_\phi(x_s, s, \emptyset), \tag{5}$$

$$x_t = \sqrt{\alpha_t}\left(\frac{x_s - \sqrt{1-\alpha_s}\epsilon_\phi(x_s, s, \emptyset)}{\sqrt{\alpha_s}}\right) + \sqrt{1-\alpha_t}\epsilon_\phi(x_s, s, \emptyset). \tag{6}$$

Then, we can integrate DDIM inversion and DDIM denoising with the same step size. We define the naive objective of 3D distillation as follows:

$$\mathcal{L}_{\text{TSM}}(\theta) := \mathbb{E}_{t,c}\left[\omega(t)||\epsilon_\phi(x_t, t, y) - \epsilon_\phi(x_\mu, \mu, \emptyset)||^2\right], \tag{7}$$

where $x_t$ and $x_\mu$ is generated through DDIM inversion from $x_0$. Following (Liang et al., 2023), the gradient of TSM loss over $\theta$ is:

$$\nabla_\theta\mathcal{L}_{\text{TSM}}(\theta) := \mathbb{E}_{t,c}\left[\omega(t)(\epsilon_\phi(x_t, t, y) - \epsilon_\phi(x_\mu, \mu, \emptyset))\frac{\partial x}{\partial \theta}\right]. \tag{8}$$

The optimization goal of TSM is to maintain the consistency of $x_0$ updates as much as possible to reduce the error introduced by DDIM inversion. Since TSM uses the same noise latent during the inversion process, its cumulative error is relatively small. In other words, both $x_\mu$ and $x_t$ come from $\hat{x}_0$ approximated by $x_s$ at time step $s$. The algorithm flow of TSM is shown in the Algorithm 1. Among them, the blue part marks the differences from ISM. For the workflow please refer to Appendix A.2.

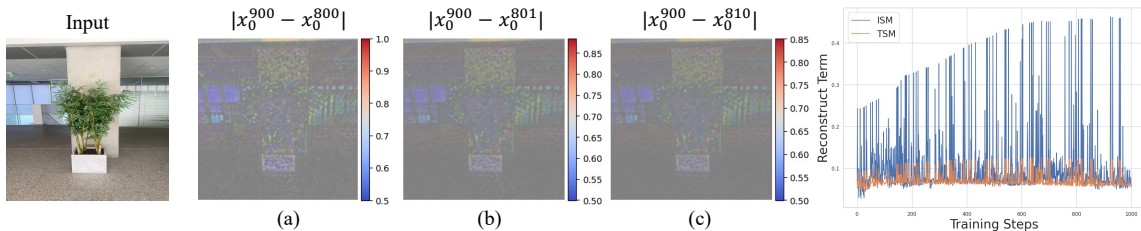

Figure 3: **The error of approximated** $x_0$. (a) is the error of x0 approximated by ISM at time step 900 and time step 800, (b) is the error of x0 approximated by TSM at time step 900 and time step 801, and (c) is the error of x0 approximated by TSM at time step 900 and time step 810.

### 3.3 COMPARISON WITH ISM

**ISM is a special case of TSM.** Theoretically, the optimization objectives of ISM and TSM are the same, however, TSM considers optimization steps that are closer together compared to ISM. Specifically, we can regard ISM as a special case of TSM. Between time steps $s$ and $t$, we choose any time step $\mu = \gamma(t - s) + s$, where $\mu = s$ if and only if $\gamma = 0$. Therefore, our hypothesis is confirmed, and ISM is indeed a special case of TSM when $\mu = s$.

**TSM has smaller error.** We conducted a 2D toy experiment. First, $x_t$ is denoised by the inverse DDIM process, and then the approximate value $\hat{x}_0$ at that time step is predicted and calculated. As shown in the figure, the error of ISM (Figure 3 (a)) is significantly larger than that of our TSM (Figure 3 (b)). It should be noted that if 1 time step is not added, TSM will degenerate into ISM. In Figure 3 (c), after 10 time steps are added, the change in error is not significant. Theoretically, at this time, there is no accumulated error between the two in the previous iteration process, and these errors mainly come from the randomness of the diffusion model. In addition, when $\mu = t$, our TSM will degenerate into CSD (Yu et al., 2023) with denoising by the inverse DDIM process. On the far right of the Figure 3, we also show the gradients of the reconstruction terms of ISM and TSM in 3D asset optimization, showing that the gradient of our method is more stable. We also provide intuitive proof in Appendix A.5.

### 3.4 THE ABNORMAL GRADIENT FROM ADVANCED PIPELINE

Previous methods have shown that increasing rendering resolution and training batch size can significantly improve visual quality. Although increasing the resolution of rendering can significantly improve the visual quality, most text-to-3D generation methods mainly use guidance based on Stable Diffusion 2.1 and only support $512 \times 512$ resolution. Due to the impact of low resolution, local details are still blurred. Consequently, we experimented with using Stable Diffusion XL as guidance, which supports $1024 \times 1024$ resolutions. The more advanced model Stable Diffusion XL has a different architecture from the previous one, and the VAE of this model is unstable. Although it has a certain impact on NeRF-based methods, it is not serious. However, this instability poses significant challenges for methods that employ 3D Gaussian splatting. In 3D Gaussian splatting, the reliability of operations like copying and deleting point clouds is heavily dependent on gradient stability. If the average positional gradient $g$ of the Gaussian view space exceeds a preset threshold, regions with under- or over-reconstruction of color $c$ and depth $d$ are intensively corrected. SDXL gradients are usually large and unstable, and high average gradient values in this case may cause normal areas to still be densified. An intuitive method is leveraging the gradient clip technical to handle this issue, previous related work (Pan et al., 2024) has explored for NeRF-based method, which is not very suitable for 3D Gaussian splatting. Thus, we propose an improved gradient clip method for 3D Gaussian splitting. Specifically, we still use the (Pan et al., 2024) method for gradient clipping of color $c$. For depth $d$, we calculate its scaling factor independently for each depth element and perform pixel-by-pixel pruning. The pruning gradient of depth $\hat{g}_d$

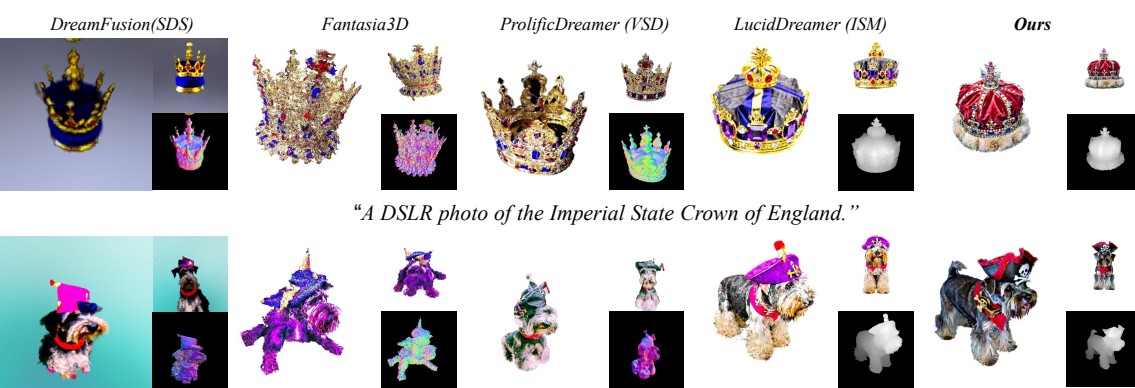

*DreamFusion(SDS)*     *Fantasia3D*     *ProlificDreamer (VSD)*     *LucidDreamer (ISM)*     ***Ours***

"*A DSLR photo of the Imperial State Crown of England.*"

"*A DSLR photo of a Schnauzer wearing a pirate hat .*"

Figure 4: **Comparison with state-of-the-art baseline methods in text-to-3D generation.** Experimental results show that our method can generate 3D content that is more consistent with input text prompts and has more detailed details. All results of this work are generated on a single A100 GPU. Please zoom in to see more details.

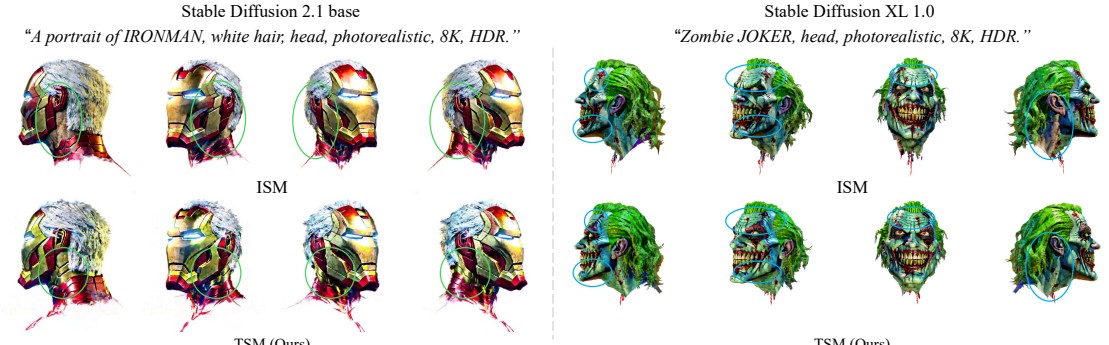

Stable Diffusion 2.1 base
"*A portrait of IRONMAN, white hair, head, photorealistic, 8K, HDR.*"

Stable Diffusion XL 1.0
"*Zombie JOKER, head, photorealistic, 8K, HDR.*"

ISM

ISM

TSM (Ours)

TSM (Ours)

Figure 5: **Comparison with the generation results of different stable diffusion models.** Compared with ISM, our TSM performs better in the clarity and consistency of local details. Please zoom in to see the circled region for more details.

can be expressed as:

$$\hat{\boldsymbol{g}}_d = \boldsymbol{g}_d \cdot \min\left(\frac{s}{|\boldsymbol{g}_d|}, c\right),\tag{9}$$

where $\boldsymbol{g}_d$ is the gradient of depth, $s$ is scale of Gaussian and $c$ is the threshold. We can ensure that the updated direction of the depth gradient remains unchanged and has no effect on the gradient of the normal region.

## 4 EXPERIMENTS

### 4.1 QUALITATIVE RESULTS

**Text-to-3D Generation.** We show the generated results of Dreamer XL in Figure 1. The results show that Dreamer XL is capable of generating high-quality 3D content accurately based on the input text, and it performs exceptionally well in producing realistic and complex appearances, effectively avoiding common issues such as excessive smoothing or oversaturation. For example, it can finely reproduce the texture

Table 1: **Quantitative evaluation.** We compare with recent text-to-3D conversion methods. CLIP-score is used to measure the alignment between text and 3D content, while A-LPIPS is used to evaluate the degree of artifacts caused by inconsistencies in 3D content.

| Methods | CLIP-Score ↑ | | A-LPIPS ↓ | |
|---|---|---|---|---|
| | CLIP-L/14 | OpenCLIP-L/14 | VGG | Alex |
| DreamFusion | 0.232 | 0.165 | 0.081 | 0.080 |
| Fantasia3D | 0.233 | 0.207 | 0.077 | 0.082 |
| ProlificDreamer | 0.255 | 0.221 | 0.178 | 0.103 |
| LucidDreamer | 0.278 | 0.234 | 0.065 | 0.059 |
| Ours (SD 2.1) | 0.281 | 0.243 | 0.059 | 0.053 |
| **Ours (SDXL)** | **0.297** | **0.312** | **0.052** | **0.041** |

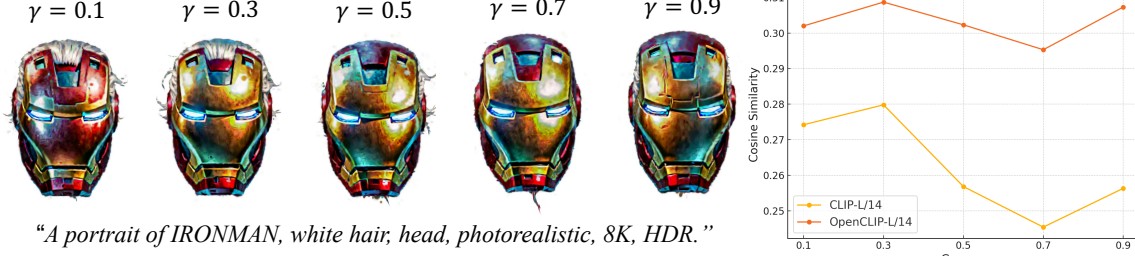

$\gamma = 0.1$  $\gamma = 0.3$  $\gamma = 0.5$  $\gamma = 0.7$  $\gamma = 0.9$

*"A portrait of IRONMAN, white hair, head, photorealistic, 8K, HDR."*

Figure 6: **Ablation** on offset rate. $\gamma = 0.3$ achieves optimal visual quality and ensures high consistency between the generated results and the original text.

details of objects like teacups. Moreover, our framework can generate objects that are close to reality and create imaginary ones. This flexibility offers possibilities for various application scenarios. **Comparison with State-of-the-Art Methods.** We compare our approach with four state-of-the-art text-to-3D baselines: DreamFusion (Poole et al., 2022) proposes Score Distillation Sampling (SDS) leveraging a pre-trained 2D text-to-image diffusion model for text-to-3D synthesis; Fantasia3D (Chen et al., 2023) disentangle geometric and appearance attributes to simulate real-world physical environments; ProlificDreamer (Wang et al., 2024) introduces Variational Score Distillation (VSD), a particle-based variational framework to address issues of oversaturation, oversmoothing, and low diversity; LucidDreamer (Liang et al., 2023) introduces Interval Score Matching (ISM), utilizing deterministic diffusion trajectories and interval-based score matching to alleviate oversmoothing problems, and employs a 3D Gaussian splatting for 3D representation. The comparison results are shown in Figure 4. The results generated by our method are significantly clearer than other baseline results. For example, the crown shows a more precise geometric structure and a more realistic color, and the Schnauzer's hair texture and overall body shape show obvious advantages. We can observe that our method significantly outperforms existing methods in both visual quality and consistency.

**Comparison with ISM in detail.** As shown in Figure 5, we show the generation results of ISM and TSM using the same prompt on different stable diffusion models. In *Iron Man*, it can be seen that the ISM has significant inconsistencies on the left and right sides of the neck, while our TSM maintains consistency in this region. In *Joker*, the ISM has shallower wrinkles on the head compared to our TSM, which is due to the averaging effect caused by error accumulation. Furthermore, ISM also shows significant inconsistency in the neck region.

## 4.2 QUANTITATIVE RESULTS

Currently, there are no standardized evaluation metrics specifically dedicated to text-to-3D. This is primarily due to the subjective nature of the task and the presence of multiple dimensions that are difficult to quantify. To maintain consistency with existing text-to-3D evaluation methods, we adopt CLIP-based metrics for quantitative analysis. Specifically, we employ variants of the CLIP model, including OpenCLIP ViT-L/14 and CLIP ViT-L/14, to calculate the average CLIP score between the text and its corresponding 3D render. Furthermore, considering the importance of view consistency, we follow previous work calculating A-LPIPS to determine view consistency, quantifying visual artifacts caused by view inconsistency through calculating the average LPIPS score between adjacent 3D scene images. We adopt A-LPIPS as an alternative metric to quantify view consistency, and present it alongside the CLIP scores in our report.

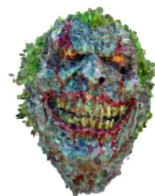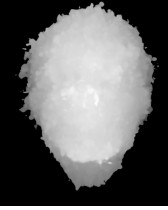

w/o gradient clipping

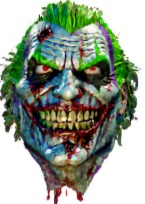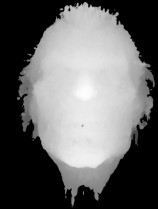

w/ gradient clipping

Figure 7: **Ablation** on pixel-by-pixel gradient clipping.

Consistent with the qualitative results, we compare our method with four state-of-the-art text-to-3D methods. Compared with the current best-performing LucidDreamer, our method improves CLIP-Score (CLIP-L/14) and CLIP-Score (OpenCLIP-L/14) by 6.83% and 33.33% respectively. At the same time, our method reduces the performance by 20.00% and 30.51% respectively on the A-LPIPS (VGG) and A-LPIPS (Alex) evaluation metrics, showing significant advantages in image authenticity and visual consistency. Overall, these results highlight the superior performance of our approach in terms of image quality and text consistency.

### 4.3 ABLATION STUDY

**Ablation on offset rate $\gamma$.** We investigate the impact of the offset rate $\gamma$ on the generated results (Figure 6), and the best results are achieved when $\gamma$ is set to 0.3. If $\gamma$ is set too low, it will result in a loss of color details; if it is set too high, it may destroy the consistency between the generated results and the text. That is, when $\mu$ is too close to $s$, the average effect is too heavy, and its effect is similar to that of ISM. When $\mu$ is too close to $t$, although the cumulative error is reduced, the updated gradient will become very small, easily causing the model to fall into a local optimum. However, for simple scenes, the results are best when $\mu$ is close to $t$, and the analysis in Appendix A.4.

**Ablation on pixel-by-pixel gradient clipping.** As shown in Figure 7, when pixel-by-pixel gradient clipping is not applied, the instability of the gradients causes abnormal splitting and duplication in normal areas, filling the depth map with noise and making it rough and uneven, thus making the entire facial appearance abnormal. However, after applying pixel-by-pixel gradient clipping, it is clearly observed that the depth map becomes smoother, the texture returns to normal, and the normal facial features are displayed. This comparison demonstrates the effectiveness of our method. For more ablation please refer to Appendix A.4.

## 5 CONCLUSION

In this work, we investigate the inconsistency problem produced by ISM during the generation of 3D results. To alleviate this problem, we introduce TSM, which leverages dual paths to reduce error accumulation and thereby improve inconsistency. In addition, to simplify the generation of a high-resolution training process, we adopt SDXL as guidance and propose a pixel-by-pixel gradient clipping method to alleviate the abnormal splitting of normal regions in 3D Gaussian splatting caused by SDXL gradient instability. Our experimental results demonstrate that our method can effectively generate high-resolution, high-quality 3D results.

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

## A APPENDIX

### A.1 IMPLEMENTATION DETAILS

During the optimization process, we train the model for 2500 iterations. To optimize the 3D Gaussian model, we set the learning rates for opacity, scaling, and rotation to 0.05, 0.005, and 0.001 respectively. Furthermore, the learning rate of the camera encoder is set to 0.001. During training, RGB images and corresponding

depth maps from 3D Gaussians are used for rendering. Gaussian densification and pruning processes are performed between 100 and 1500 iterations. We select the publicly available stable diffusion of text to images as a guidance model and choose the checkpoint of Stable Diffusion XL[1]. The guidance scale is 7.5 for all diffusion guidance models.

## A.2 THE WORKFLOW OF ISM AND TSM

As shown in Figure 8, we present ISM and our TSM workflow.

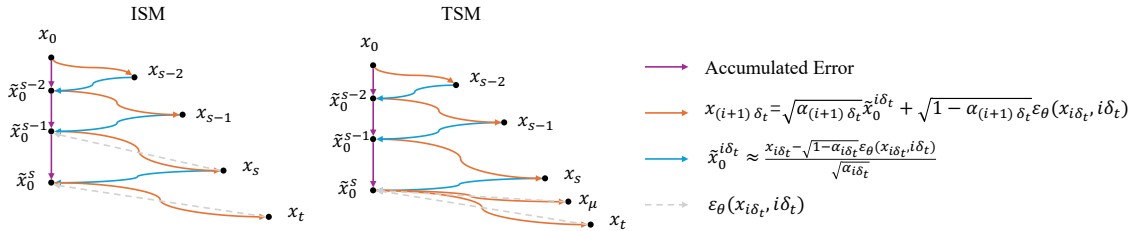

Figure 8: The Workflow of ISM and TSM.

## A.3 MORE QUALITATIVE RESULTS

As shown in Figure 9, we present additional generation results. It can be observed that our Dreamer XL is capable of generating 3D models that are visually high-quality, closely approximate reality, and maintain good consistency.

## A.4 MORE ABLATION STUDY

**Ablation on offset rate $\gamma$ for simple scene**  Here, we provide more ablation on offset rate $\gamma$ for simple scene. As shown in Figures 10 and 11, if $\gamma$ is set too low, the similarity between text and images decreases, and the images appear unrealistic. Raising $\gamma$ can mitigate the problem of accumulated errors, but it also leads to a loss of some details.

**Ablation on pixel-by-pixel gradient clipping**  Here, we present additional result in Figures 12 and 13. In Figure 12, once gradient clipping is not applied, the generated 3D model tends to be darker in color and lacks realism. In Figure 13, without gradient clipping, the generated objects appear more blurry, and it is evident from the depth map that there are numerous noise and spikes. As shown in Figure 14 for SD 2.1, there are some minor changes but they are not obvious.

## A.5 INTUITIVE PROOF OF CONSISTENCY IN TSM

Here, we provide an intuitive argument to illustrate the underlying principles. Consider the DDIM inversion process, where cumulative errors lead to deviations in the approximation $\hat{x}_0$. TSM minimizes the error accumulation inherent in DDIM inversion by starting from the same starting point. TSM maintains a stable approximation of $\hat{x}_0$ across different perspectives. This synchronization reduces the variance in feature

---

[1]https://huggingface.co/stabilityai/stable-diffusion-xl-base-1.0

*"A Gundam model, with detailed panel lines and decals."*

*"An action figure of Iron Man, Marvel's Avengers, HD."*

*"A pillow."*

*"A DSLR photo of a Cream Cheese Donut."*

*"a red apple."*

*"A ripe strawberry."*

*"a DSLR photo of a cat wearing armor."*

*"A plate piled high with chocolate chip cookies."*

*"A durian, 8k, HDR."*

*"Marble bust of Theodoros Kolokotronis."*

*"A DSLR photo of A Rugged, vintageinspired hiking boots with a weathered leather finish, best quality, 8K, HD."*

*"a DSLR photo of A very beautiful tiny human heart organic sculpture made of copper wire and threaded pipes, very intricate, curved, Studio lighting, high resolution."*

*"A 3D model of an adorable cottage with a thatched roof."*

*"A DSLR photo of A Stylish Air Jordan shoes, best quality, 8K, HD."*

Figure 9: More results generated by our Dreamer XL framework. Please zoom in for details.

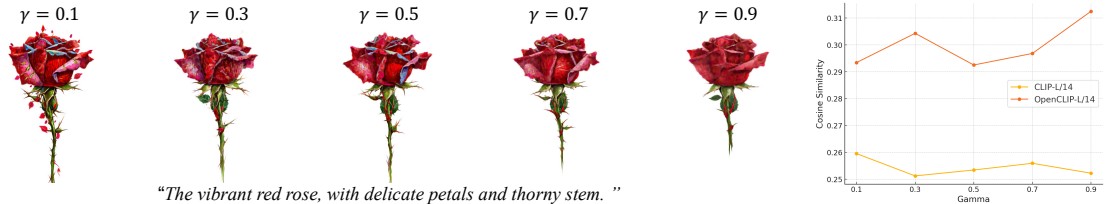

*"The vibrant red rose, with delicate petals and thorny stem."*

Figure 10: **More Ablation** on offset rate $\gamma$ for simple scene.

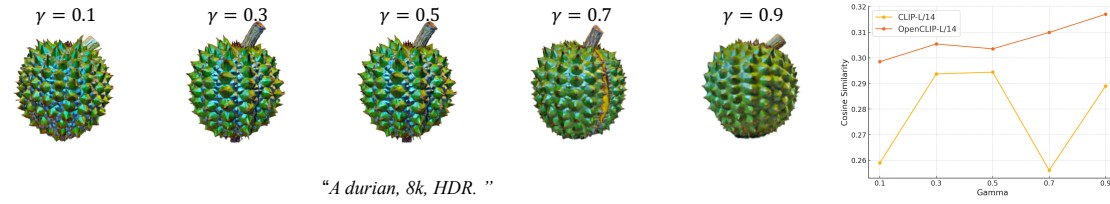

Figure 11: **More Ablation** on offset rate $\gamma$ for simple scene.

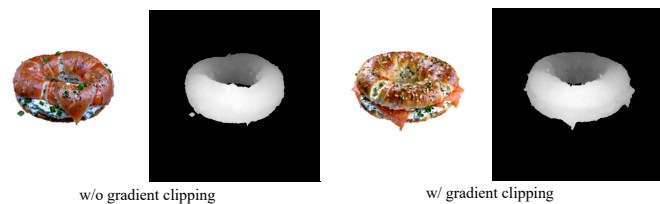

Figure 12: **More Ablation** on pixel-by-pixel gradient clipping.

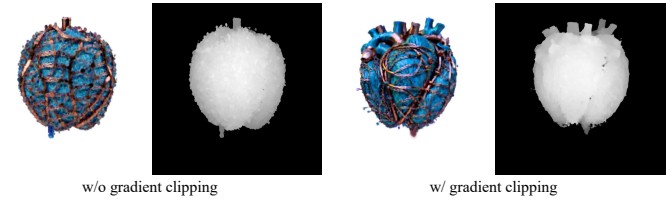

Figure 13: **More Ablation** on pixel-by-pixel gradient clipping.

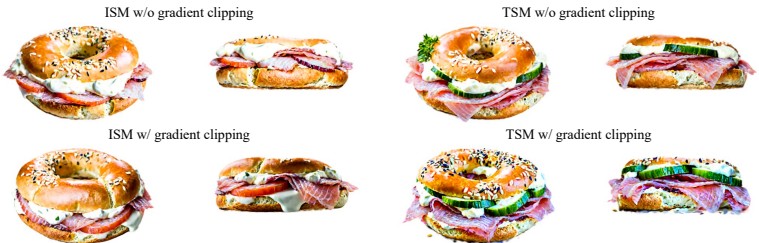

Figure 14: textbfMore Ablation on pixel-by-pixel gradient clipping for SD 2.1.

representations and stylistic attributes, leading to more consistent optimization outcomes. Next, we discuss in detail the optimization objectives and error accumulation of ISM (single-path optimization) and TSM (dual-path optimization).

For ISM, the optimized goal is the minimum predicted noise from noise latent $x_s$ and $x_t$. The noise latent can be obtained from Equation (2) and then using the pre-trained stable diffusion model to predict noise. During

the inversion process of DDIM, $x_s$ is approximated by the prediction noise at $s - 1$, and $x_t$ is approximated by the prediction noise at $s$. For simplicity, we can regard the optimization goal of ISM as minimizing noise latent, thus the accumulated error generated by ISM during the optimization process can be expressed as:

$$x_t - x_s = \sqrt{\alpha_t} \left( \frac{x_s - \sqrt{1 - \alpha_s}\epsilon_\phi(x_s, s, \emptyset)}{\sqrt{\alpha_s}} \right) + \sqrt{1 - \alpha_t}\epsilon_\phi(x_s, s, \emptyset)$$
$$- \sqrt{\alpha_s} \left( \frac{x_{s-1} - \sqrt{1 - \alpha_{s-1}}\epsilon_\phi(x_{s-1}, s - 1, \emptyset)}{\sqrt{\alpha_{s-1}}} \right) + \sqrt{1 - \alpha_s}\epsilon_\phi(x_{s-1}, s - 1, \emptyset) \quad (10)$$

For our TSM, we can also give similar expresses as:

$$x_t - x_\mu = \sqrt{\alpha_t} \left( \frac{x_s - \sqrt{1 - \alpha_s}\epsilon_\phi(x_s, s, \emptyset)}{\sqrt{\alpha_s}} \right) + \sqrt{1 - \alpha_t}\epsilon_\phi(x_s, s, \emptyset)$$
$$- \sqrt{\alpha_\mu} \left( \frac{x_s - \sqrt{1 - \alpha_s}\epsilon_\phi(x_s, s, \emptyset)}{\sqrt{\alpha_s}} \right) + \sqrt{1 - \alpha_\mu}\epsilon_\phi(x_s, s, \emptyset) \quad (11)$$

Compared with ISM, which is optimized for a single path, TSM is optimized for a dual path starting from the same noise latent. We show that the accumulated error is smaller in TSM, as shown below.

**Definition 1** (Accumulated Error in DDIM Reverse Process) The additional accumulated error during the DDIM reverse process is the difference between two noise estimates from consecutive iterations. Let $x_{i+1}$ and $x_i$ represent the noise approximate at iterations $i + 1$ and $i$ respectively. The additional accumulated error $\eta$ is given by:

$$\eta = x_{(i+1)\delta_t} - x_{i\delta_t} \quad (12)$$

where $\delta_t$ denotes the time step size in the iteration process.

**Proposition 1** (TSM has lower additional accumulated error) *Consider three timesteps $0 < s < \mu < t$, where $\mu$ is defined as $\mu = \gamma(t - s) + s$, with $\gamma \in [0, 1]$. If the additional accumulated error of ISM, $\eta_{ISM}$, is higher than additional accumulated error of TSM, $\eta_{TSM}$, then, we have:*

$$\eta_{\text{ISM}} > \eta_{\text{TSM}} \Rightarrow (x_t - x_s) - (x_t - x_\mu) > 0 \Rightarrow x_\mu - x_s > 0, \quad (13)$$

*Proof:* Assume $x_\mu \leq x_s$. Utilizing Equation (10) and Equation (11), we aim to demonstrate that:

$$\sqrt{\alpha_\mu} \left( \frac{x_s - \sqrt{1 - \alpha_s}\epsilon_\phi(x_s, s, \emptyset)}{\sqrt{\alpha_s}} \right) + \sqrt{1 - \alpha_\mu}\epsilon_\phi(x_s, s, \emptyset)$$
$$\leq \sqrt{\alpha_s} \left( \frac{x_{s-1} - \sqrt{1 - \alpha_{s-1}}\epsilon_\phi(x_{s-1}, s - 1, \emptyset)}{\sqrt{\alpha_{s-1}}} \right) + \sqrt{1 - \alpha_s}\epsilon_\phi(x_{s-1}, s - 1, \emptyset). \quad (14)$$

Where $\alpha$ represents a set of timesteps with a strictly monotonically reducing noise schedule, hence $\alpha_{s-1} > \alpha_s > \alpha_\mu$. The first term can be written as $\sqrt{\alpha_\mu}\tilde{x}_0^s$ and $\sqrt{\alpha_s}\tilde{x}_0^{s-1}$, where the coefficients ensure the correct scale. The latter term $\tilde{x}_0^s$ iterates one more time than $\tilde{x}_0^{s-1}$, thus accumulating one more iteration of error. Therefore, $\tilde{x}_0^s > \tilde{x}_0^{s-1}$, which is contrary to the assumption. Considering the $\epsilon_\phi$ terms, due to the inversion process of DDIM, $\epsilon_\phi(x_s, s, \emptyset) \approx \epsilon_\phi(x_{s-1}, s - 1, \emptyset)$ and $\sqrt{1 - \alpha_\mu} > \sqrt{1 - \alpha_s}$. This also contradicts the assumption. Therefore, our initial assumption $x_\mu \leq x_s$ must be false. Consequently, we conclude that $x_\mu > x_s$, thus proving the theorem.

## A.6   LIMITATIONS

While our method can generate high-quality and relatively realistic 3D models, qualitative results show a limitation in our approach regarding light handling. Specifically, we have observed anomalous blue reflections in many scenes. Through our experiments, we have identified this problem as primarily caused by our use of SDXL. When SDXL is applied, the blue channel values in rendered images tend to be large, resulting in numerous areas exhibiting abnormal blue hues after normalization. Despite our attempts, including parameter adjustments and different normalization methods, we have yet to find a viable solution. We speculate that this may be attributed to the gradient or training data of SDXL. Additionally, it's worth noting that while our work aims to enhance the quality of generated models, it may inadvertently contribute to the advancement of deepfake technology.

