# OpenReview forum: "Dreamer XL: Towards High-Resolution Text-to-3D Generation via Trajectory Score Matching"
_ICLR.cc/2025/Conference — ICLR 2025 Conference Withdrawn Submission_

### Official Review · Reviewer_ip1q · 2024-10-20

**Soundness:** 1
**Presentation:** 2
**Contribution:** 1
**Rating:** 3
**Confidence:** 4

**Summary:**

This paper proposes Trajectory Score Matching (TSM) to address the accumulated error of DDIM inversion used in ISM. Specifically, the authors added an additional inversion step to reduce the error in the loss to reduce the error. The authors also proposed depth gradient clipping to stablize the training.

**Strengths:**

1. The paper clearly distinguishes the differences between the proposed TSM and ISM methods.
2. The introduction of depth gradient pruning could be a valuable engineering approach for practical applications.

**Weaknesses:**

1. The theoretical foundation presented in this paper is not robust and sound. Specifically, the naive distillation objective in Eq. (7) lacks intuitiveness and requires further explanation in the context of this paper. Additionally, the proof of Proposition 1 appears to be incorrect, as it directly compares vectors (instead of their norm lengths). Without a complete proof, it is difficult to accept that TSM can effectively reduce the "errors." Figure 3 does not clarify the differences, and the main benefit of TSM seems to be the reduction of gradient variance, which is not well justified by quantitative measurements.

2. The motivation for reducing prediction error is inadequately explained. The inconsistencies shown in Figure 2 are unconvincing, as both ISM and TSM produce mostly identical results with only minor differences. The advantages of TSM in terms of reduced error are not clearly demonstrated, and when the difference between $\boldsymbol{\epsilon} _\phi(\boldsymbol{x} _t)$ and $\boldsymbol{\epsilon} _\phi(\boldsymbol{x} _\mu)$ is small, the impact of classifier-free guidance may be amplified. Moreover, if the goal is merely to reduce error, setting $t = \mu$ would eliminate the "error" to 0. Reducing the tunable parameter $\delta_T = t - s$ in ISM could also achieve error reduction. An experiment comparing ISM with TSM using the same $t - \mu = \delta_T$ is necessary.

3. There is a lack of fair comparison. The comparisons presented appear unfair, as the authors use two different base diffusion models in Table 1, with TSM using SDv2.1 showing only minimal improvements over ISM. This improvement may also originate from the increase in the impact of CFG as noted in weakness 2. As shown in figure 5, the differences between ISM and TSM is also minor. A fair comparison utilizing the same diffusion model is essential.

**Questions:**

1. Regarding weakness 1, did the authors account for CFG when computing the "reconstruction term" in Figure 3?

2. Regarding weakness 3, what are the base diffusion models used for the baselines?

3. What exactly does the A-LPIPS metric in Section 4.2 refer to? This metric does not seem common in the field. The authors should cite relevant previous work clearly and provide a detailed explanation of this metric.

4. How many prompts and seeds did the authors experiment with in Section 4.2? The authors should report these numbers in the experimental details. Conducting experiments with at least 25 samples would provide a more robust measurement.

---

### Official Review · Reviewer_RD4o · 2024-11-01

**Soundness:** 2
**Presentation:** 3
**Contribution:** 2
**Rating:** 5
**Confidence:** 3

**Summary:**

This paper targets optimization based text-to-3D generation, aiming to resolve the inconsistent images generated by 2D DDPM used for supervising the training of 3D neural parameters in SDS. Specifically, it introduces Trajectory Score Matching (TSM), a method designed to improve upon the Interval Score Matching (ISM) technique introduced in LucidDreamer, addressing the inconsistent supervision issues present in SDS. Compared to ISM, TSM adds an additional noise latent on the trajectory, claiming it will help reduce the accumulated errors in the DDIM inversion. As the paper uses SDXL as the 2D diffusion prior, it additional proposes a pixel-by-pixel gradient clipping method to resolve the gradient instability issue.

**Strengths:**

The paper is well organized and clearly presented.

**Weaknesses:**

The motivation for using the dual paths appears somewhat ad-hoc.

In Fig.2, notable inconsistencies still exist for the proposed TSM, such as variations in hair, face color, and neck.

Comparisons between TSM and ISM are shown only in Figure 5, which is insufficient to demonstrate the effectiveness of TSM.

There are no video demos showcasing the generated 3D results.

**Questions:**

Besides the optimization based methods, another line of work is feed-forward generation method, which directly train 3D generative models from large-scale 3D datasets. This work focuses on the optimization based method. The related work section can be made more clear on this and be more specific.

In Figure 3, the difference between ISM and TSM appears to be primarily a matter of scale. Could you provide further analysis on this?

What kind of dataset is used for the quantitative evaluation in Table 1? It should be a commonly used benchmark, such as the T^3 benchmark.

In Table 1, what diffusion model is used by LucidDreamer? Is an eye-to-eye comparison ensured?

---

### Official Review · Reviewer_oyBX · 2024-11-01

**Soundness:** 3
**Presentation:** 3
**Contribution:** 2
**Rating:** 3
**Confidence:** 4

**Summary:**

The paper presents a promising new method to enhance the visual quality and performance of text-to-3D generation. But there are some limitations.

**Strengths:**

The paper introduces a novel Trajectory Score Matching (TSM) method to address the accumulated error issue in Interval Score Matching (ISM), which is an interesting technical innovation. The authors demonstrate the advantages of TSM in reducing accumulated error and improving the stability of model-generated paths, supported by experiments. The paper is well-structured, and the discussion of related work helps readers understand the progress in the field. The authors' approach of using Stable Diffusion XL as guidance and proposing a pixel-by-pixel gradient clipping method to address the unstable gradient issues during 3D Gaussian splatting is practical.

**Weaknesses:**

1. Although the paper proposes TSM to reduce the accumulated error in ISM, it remains to be further validated whether TSM can fully address the issue of accumulated errors, especially when dealing with more complex 3D models where the impact of accumulated errors might be more significant.
2. This paper use the powerful base model SDXL for comparison with the SD model-based methods is unfair because this paper is employing a stronger prior base model.
3. This paper utilizes the SDXL model, which is significantly slower than SD2-1 in 2D generation, suggesting it would be extremely slow for 3D generation. The paper does not discuss comparative experiments on time consumption.
4. The paper briefly mentions limitations, such as issues with light handling, but does not provide a detailed analysis of when the model fails or its limitations in complex scenarios.

**Questions:**

Please refer to Weaknesses.

---

### Official Review · Reviewer_yhpP · 2024-11-04

**Soundness:** 2
**Presentation:** 2
**Contribution:** 2
**Rating:** 3
**Confidence:** 4

**Summary:**

This paper introduces TSM, which they argue is an improved for of Interval Score Matching (ISM) introduced by LucidDreamer, in which DDIM inversion process is leveraged to generate two paths for score distillation. The paper also introduces a pixel-by-pixel gradient clipping method, which according to their statement alleviates abnormal gradient from advanced pipeline.

**Strengths:**

- The paper is well-written and easy to follow.
- The paper offers sound theoretical background on trajectory score matching, in which they modify ISM so that instead of conducting matching between intervals of a trajectory, two trajectories starting from same latent of a certain timestep are compared and used as a loss for gradient distillation into the 3D scene. Their explanation that this reduces error accumulation from DDIM inversion is intuitive and makes sense.

**Weaknesses:**

- **Unclear motivation.** This paper's states that the main problem in previous text-to-3D framework (ISM) is "pseudo ground truth inconsistency" that derives from accumulation of error within the DDIM inversion process. However, the term "pseudo ground truth" is never clearly defined within the paper, failing to convince the reader of the necessity of their task.
- **Lack of justification for the method.** Furthermore, the paper fails to clearly deliver critical failure modes caused by ISM due to the error accumulation caused by DDIM inversion: in order for its motivation to be meaningful, cases of failure or degradation resulting from ISM-based method, from which TSM improves upon, must be given. Such comparisons are crucial to the credibility of the paper, but comparison of results with those of ISM are not given within the paper, making the reader question the necessity of their contribution.
- **Lack of experiment.** The paper gives two contributions, TSM and pixel-level gradient clipping, but I believe the ablation studies to be lacking, with the results given in Appendix showing marginal performance improvement at best. In fact, most qualitative results in the paper show very little to marginal improvement from the results given in LucidDreamer (ISM).
- **Clarification on performance gain from using SDXL.** Table 1 shows quantitative evaluation and comparison to previous methods: however, major improvement in performance in the given table seems to be resulting from changing of diffusion model from SD 2.1 to SDXL, which is given and not related to any form of contribution stated in the paper. I believe all compared works utilize SD 2.1 or earlier version of Stable Diffusion, and in this setting TSM seems to show very marginal quantitative improvement.

**Questions:**

- Please give clear examples of failure modes caused by ISM, and clear improvement from the same setting (prompt, seed, SD version) using TSM, as well as improved ablation studies and more abundant comparison to other SDS-based text-to-3D generation works.

---

### Note · Authors · 2024-11-13

I have read and agree with the venue's withdrawal policy on behalf of myself and my co-authors.